# Assessing the extent and public health impact of bat predation by domestic animals using data from a rabies passive surveillance program

Amy G. Wilson[1]*, Christine Fehlner-Gardiner[2], Scott Wilson[1,3], Karra N. Pierce[4], Glenna F. McGregor[5], Catalina González[1], Tanya M. J. Luszcz[6]

1 Department of Forest and Conservation Sciences, University of British Columbia, Vancouver, British Columbia, Canada, 2 Centre of Expertise for Rabies, Ottawa Laboratory Fallowfield, Canadian Food Inspection Agency, Ottawa, Ontario, Canada, 3 Wildlife Research Division, Science and Technology Branch, Environment and Climate Change Canada, Delta, British Columbia, Canada, 4 Wildlife Center of Virginia, Waynesboro, Virginia, United States of America, 5 Animal Health Centre, British Columbia Ministry of Agriculture, Abbotsford, British Columbia, Canada, 6 Canadian Wildlife Service, Environment and Climate Change Canada, Penticton, British Columbia, Canada

* amy.wilson@ubc.ca

**Data Availability Statement:** All data are in the manuscript and/or Supporting information files.

## Abstract

Domestic animals can serve as consequential conveyors of zoonotic pathogens across wildlife-human interfaces. Still, there has been little study on how different domestic species and their behaviors influence the zoonotic risk to humans. In this study, we examined patterns of bat encounters with domestic animals that resulted in submission for testing at the rabies laboratories of the Canadian Food Inspection Agency (CFIA) during 2014–2020. Our goals were specifically to examine how the number of bats submitted and the number of rabies positive bats varied by the type of domestic animal exposure and whether domestic cats were indoor or free-roaming. The CFIA reported 6258 bat submissions for rabies testing, of which 41.5% and 8.7% had encounter histories with cats and dogs, respectively. A much smaller fraction of bat submissions (0.3%) had exposure to other domestic animals, and 49.5% had no domestic animal exposure. For the bat submissions related to cats, and where lifestyle was noted, 91.1% were associated with free-roaming cats and 8.9% with indoor cats. Model results indicated the probability of a rabies-positive bat was the highest with a history of dog association (20.2%), followed by bats with no animal exposure (16.7%), free-roaming cats (6.9%), cats with unspecified histories (6.0%) and the lowest probability associated with non-free-roaming (indoor) cats (3.8%). Although there was lower rabies prevalence in bats associated with cats compared to dogs, the 4.8 fold higher number of cat-bat interactions cumulatively leads to a greater overall rabies exposure risk to humans from any free-roaming outdoor cats. This study suggests that free-roaming owned cats may have an underappreciated role in cryptic rabies exposures in humans and as a significant predator of bats. Preventing free-roaming in cats is a cost-effective and underutilized public health recommendation for rabies prevention that also synergistically reduces the health

**Funding:** Funding provided by Environment and Climate Change Canada (AW). The funders had no role in study design, data collection and analysis, decision to publish, or preparation of the manuscript.

**Competing interests:** The authors have declared that no competing interests exist.

burden of other feline-associated zoonotic diseases and promotes feline welfare and wildlife conservation.

## Introduction

The wildlife-human interface has received considerable attention as a source of emerging disease and zoonotic disease transfer [1], making it essential to understand the mechanisms functioning at this interface. Although humans share more pathogens with domestic animals than wildlife [2–5], domestic animal contact with wildlife increases the risk of secondary transmission of wildlife zoonoses to humans. Disease transmission risk increases when ecosystem degradation reduces the disease resiliency of wildlife [6], and habitat encroachment places wildlife in abnormal contact situations such as live animal markets [7] or through frequent interactions with domestic animals. Frequent close contact between wildlife and free-roaming domestic animals increases the opportunity for bidirectional disease transfer [8].

Depredation of wildlife by free-roaming domestic animals (primarily cats and dogs) is a significant source of wildlife mortality at a global scale [9–11]; even conservative estimates of cat-caused mortality from a few countries are on the order of tens of billions of wildlife individuals [9, 10]. Beyond the clear conservation concerns, these depredation events are of public health significance because each domestic animal predation event equates to an opportunity for disease transmission across the wildlife-human interface. Domestic cats are a particular concern for conservation, welfare and public health in many regions because they are often allowed to roam free without supervision and, as a result, frequently interact with and kill wildlife. Free-roaming cats not only present a risk for transmitting viral, parasitic and bacterial diseases to wildlife [4, 8, 11–13], but also contracting and transporting diseases from wildlife that can be then passed on to humans [11].

There are an estimated 5.4–9.6 million free-roaming cats in Canada, yet only a few studies have examined their conservation [14] or public health consequences [15, 16]. However, there is a growing awareness of the impact of free-roaming cat predation on bat populations [17, 18], along with implications of these interactions for human rabies risk [19]. Rabies is an RNA virus from the *Lyssavirus* genus in the Rhabdoviridae family that can infect most mammalian species leading to acute, progressive fatal encephalitis [20]. There are multiple mammalian reservoirs for rabies, including domestic dogs, foxes, skunks, raccoons and bats, although in North America, only the latter four are considered rabies reservoirs [20, 21]. Estimates of the natural prevalence of rabies within North American bat populations are low and around 0.5% [22]. Nevertheless, rabies is a disease of major public health significance because of the near absolute fatality rate for untreated humans. Therefore, any bats involved in situations with an identified human transmission risk are euthanized and tested. These public health records can provide insight into the circumstances leading to these interactions, enabling the design and implementation of preventative measures for the mutual protection of humans and wildlife.

In this study, we examined how the number of bats submitted and the number of rabies positive bats varied by the type of domestic animal exposure and in relation to the lifestyle of domestic cats and whether they were indoor or free-roaming outdoors. To address these objectives, we used patterns of bat submissions for rabies virus testing in Canada as a model system. Our overall hypothesis was that free-roaming cats act as consequential transport hosts for zoonotic disease across the wildlife-human interface because of their frequent interaction with wildlife while roaming unsupervised. Under this hypothesis, we first predicted that bat

submissions in Canada would more commonly be associated with cats than dogs because of the higher rates of free-roaming in cats than dogs. We also predicted that within cats, those associated with a free-roaming outdoor lifestyle would have a higher probability of interacting with rabies-positive bats compared to cats associated with an indoor lifestyle. Due to the less frequent free-roaming of dogs in North America, we hypothesized that dog-bat interactions would primarily be incidental and occur with grounded bats already incapacitated by a predisposing injury or illness. Therefore, we predicted higher rabies prevalence in dog-associated bats relative to cats.

## Methods

Rabies testing submission data were available from the Canadian Food Inspection Agency's (CFIA) Centre of Expertise for Rabies from 2014 to 2020. Samples were submitted to the CFIA from provincial and territorial public health and animal health authorities across Canada. In general, bats are submitted to the CFIA for analysis in cases of a domestic animal or human exposure through bites, scratches, or abrasions. Samples were classified as associated with a dog, cat, livestock or another animal. Bats were classified in a single, 'no domestic animal contact' group if there was direct or indirect human contact or bats were submitted for confirmatory testing in wildlife disease surveillance programs. For each submission, information was available on the date received, municipality where the potential exposure occurred, bat species involved and the species of domestic animal potentially exposed. We also reviewed the comments section of each record to obtain any available vaccination information and the lifestyle (free-roaming or indoor) of household cats. Rabies testing utilized brain tissue in a fluorescent antibody test [23] or an immunohistochemical assay [24], with samples classified as positive, negative or unfit for testing.

### Statistical analysis

All data manipulations, analyses, and graphics used R version 4.0. R Studio Version 1.3 [25] and the tidyverse package [26]. We estimated the rabies prevalence with 95% confidence intervals using the R package epiR [27]. Samples that were classified as unfit for rabies testing were removed (n = 392) from all statistical analyses. Rabies prevalence estimates were calculated from data pooled across bat species, provinces and years for bats associated with and without domestic animal exposure.

We evaluated the factors associated with the presence of a rabies-positive bat using generalized linear mixed models (log-link function) assuming a binomial error distribution as implemented in the R package lme4 [28]. We estimated the influence of domestic companion animal exposure on the probability of a rabies-positive bat while controlling for season, bat species, year and province through comparison of three potential models. To control for potentially influential phylogenetic or ecological differences among bats species and due to disparate sampling across provinces and years, we modeled bat species, province and year as random effects. The influences of season (spring, summer, fall and winter) and companion animal exposure (dog, cat with unspecified lifestyle, cat indoor, cat outdoor, equine, other livestock and other pet) were modeled as fixed effects. Model support was based on Akaike's Information Criterion (AIC), where models with ΔAIC < 2 were considered equally plausible [6].

## Results

Across six years, ten provinces, and two territories, there were 6258 submissions for 14 species of bats (Table 1, S1 Data): Big brown bat (EPFU, *Eptesicus fuscus*), California bat (MYCA, *Myotis californicus*), fringed bat (MYTH, *M. thysanodes*), hoary bat (LACI, *Lasiurus cinereus*),

**Table 1. The total number of bats (top) and the number of rabies-positive bats relative to total samples fit for testing (bottom) submitted to the CFIA during 2014–2020 summarized by province and bats species.** Bat species codes are discussed in text.

| Species | Sps total | BC | YT | AB | SK | MB | NT | ON | QC | NB | NL | PE | NS |
|---|---|---|---|---|---|---|---|---|---|---|---|---|---|
| COTO | 12 | 12 | 0 | 0 | 0 | 0 | 0 | 0 | 0 | 0 | 0 | 0 | 0 |
|  | 1/11 | 1/11 |  |  |  |  |  |  |  |  |  |  |  |
| EPFU | 4084 | 137 | 0 | 163 | 335 | 8 | 0 | 3045 | 300 | 95 | 1 | 0 | 0 |
|  | 319/3897 | 29/132 |  | 18/151 | 45/323 | 0/8 |  | 176/2900 | 36/291 | 15/91 | 0/1 |  |  |
| LABO | 16 | 0 | 0 | 2 | 0 | 1 | 0 | 12 | 1 | 0 | 0 | 0 | 0 |
|  | 1/13 |  |  | 0/2 |  | 1/1 |  | 0/9 | 0/1 |  |  |  |  |
| LACI | 47 | 5 | 0 | 8 | 6 | 7 | 0 | 18 | 1 | 2 | 0 | 0 | 0 |
|  | 15/44 | 2/5 |  | 1/8 | 0/4 | 1/6 |  | 8/18 | 1/1 | 2/2 |  |  |  |
| LANO | 266 | 38 | 0 | 150 | 46 | 14 | 0 | 16 | 1 | 1 | 0 | 0 | 0 |
|  | 30/249 | 7/36 |  | 16/141 | 3/42 | 0/12 |  | 3/16 | 1/1 |  |  |  |  |
| MYCA | 227 | 227 | 0 | 0 | 0 | 0 | 0 | 0 | 0 | 0 | 0 | 0 | 0 |
|  | 12/206 | 12/206 |  |  |  |  |  |  |  |  |  |  |  |
| MYCI | 34 | 5 | 0 | 26 | 3 | 0 | 0 | 0 | 0 | 0 | 0 | 0 | 0 |
|  | 2/34 | 0/5 |  | 2/26 | 0/3 |  |  |  |  |  |  |  |  |
| MYEV | 54 | 52 | 0 | 2 | 0 | 0 | 0 | 0 | 0 | 0 | 0 | 0 | 0 |
|  | 4/48 | 3/46 |  | 1/2 |  |  |  |  |  |  |  |  |  |
| MYKE | 30 | 30 | 0 | 0 | 0 | 0 | 0 | 0 | 0 | 0 | 0 | 0 | 0 |
|  | 0/27 | 0/27 |  |  |  |  |  |  |  |  |  |  |  |
| MYLU | 1075 | 226 | 1 | 477 | 79 | 26 | 2 | 111 | 10 | 7 | 115 | 14 | 7 |
|  | 21/982 | 11/209 | 0/1 | 4/436 | 3/72 | 0/18 | 0/2 | 3/101 | 0/9 | 0/7 | 0/107 | 0/14 | 0/6 |
| MYSE | 66 | 24 | 0 | 34 | 4 | 0 | 1 | 1 | 0 | 0 | 0 | 1 | 1 |
|  | 3/66 | 2/24 |  | 1/34 | 0/4 |  | 0/1 | 0/1 |  |  |  | 0/1 | 0/1 |
| MYTH | 1 | 1 | 0 | 0 | 0 | 0 | 0 | 0 | 0 | 0 | 0 | 0 | 0 |
|  | 1/1 | 1/1 |  |  |  |  |  |  |  |  |  |  |  |
| MYVO | 9 | 4 | 0 | 5 | 0 | 0 | 0 | 0 | 0 | 0 | 0 | 0 | 0 |
|  | 0/9 | 0/4 |  | 0/5 |  |  |  |  |  |  |  |  |  |
| MYYU | 237 | 236 | 0 | 1 | 0 | 0 | 0 | 0 | 0 | 0 | 0 | 0 | 0 |
|  | 9/198 | 8/197 |  | 1/1 |  |  |  |  |  |  |  |  |  |
| UNID | 100 | 35 | 0 | 16 | 32 | 0 | 0 | 10 | 6 | 0 | 1 | 0 | 0 |
|  | 17/80 | 6/29 |  | 0/12 | 6/30 |  |  | 1/3 | 4/6 |  | 0/0 |  |  |

Keen's bat (MYKE, *M. keenii*), little brown bat (MYLU, *M. lucifugus*), long-eared bat (MYEV, *M. evotis*), long-legged bat (MYVO, *M. volans*), northern myotis (MYSE, *M. septentrionalis*), eastern red bat (LABO, *L. borealis*), silver-haired bat (LANO, *Lasionycteris noctivagans*), Townsend's big-eared bat (COTO, *Corynorhinus townsendii*), western small-footed bat (MYCI, *M. ciliolabrum*) and Yuma bat (MYYU, *M. yumanensis*). The numbers of bats tested differed between provinces, with the majority of tested bats originating from Ontario (50%; n = 3213), followed by British Columbia (16%; n = 1032) and Alberta (14%; n = 884) (Table 1). The bat species submitted were predominantly big brown bat (66%, n = 4084), followed by little brown bat (17%, n = 1075), silver-haired bat (4%, n = 266) and Yuma bat (4%, n = 237) (Table 1).

Overall, submitted bats had contact with cats for 41.5% of all cases (n = 2599) and with dogs for 8.7% of cases (n = 542) (Figs 1 and 2). The remaining animal exposures involved goat (n = 1), horse (n = 8), livestock (n = 1), other non-cat or dog companion pet species (n = 9) and raccoon (n = 1). Cases with no domestic animal contact comprised 49.5% of the submissions (n = 3097).

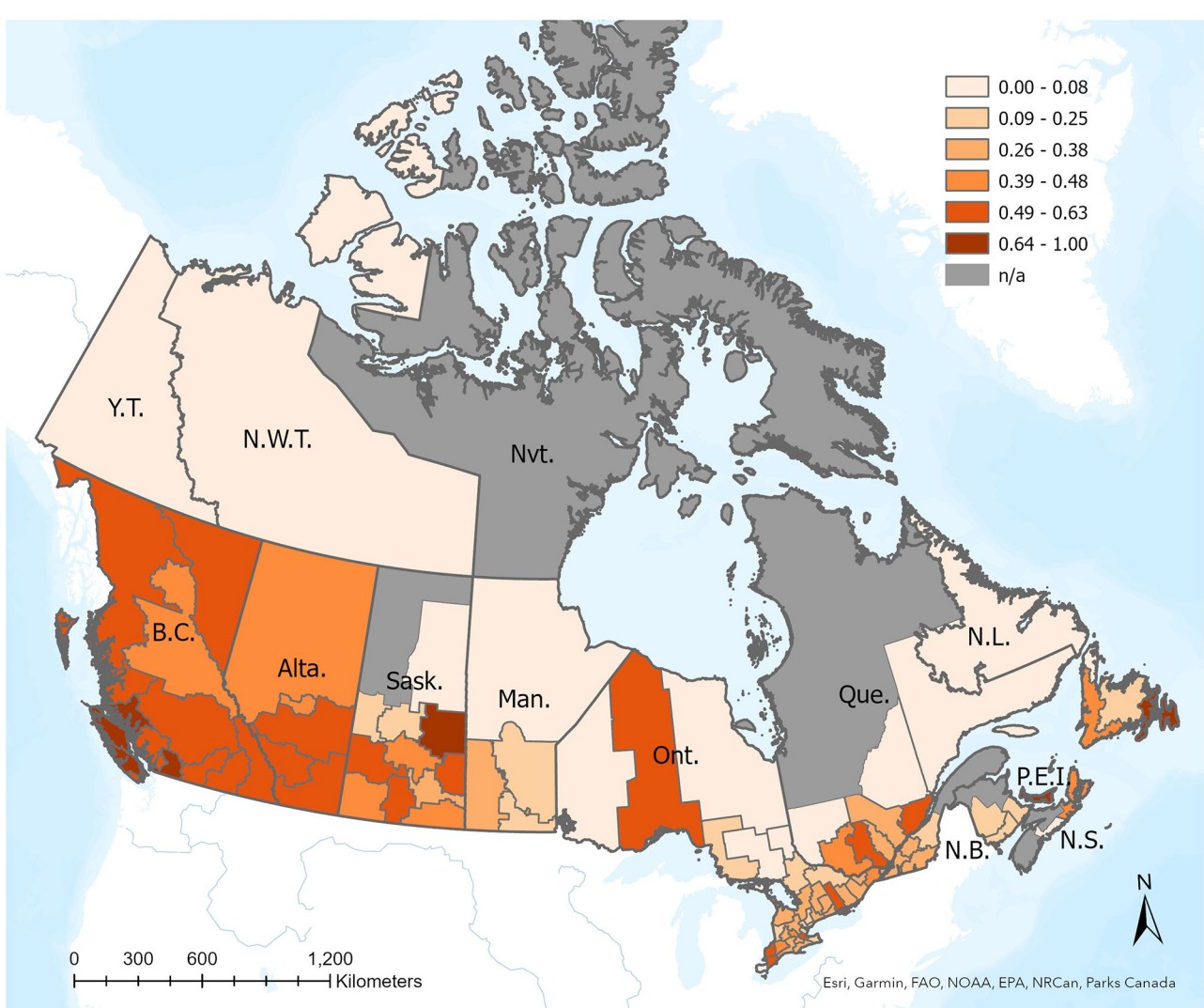

**Fig 1. Distribution of bat submission for rabies testing to CFIA during 2014–2020.** Samples are pooled across public health authorities and the percent of submission of bats that were associated with cats is shown in a gradation of orange with darker red indicating a higher percentage. Public heatlh authorities with no submissions are depicted in grey with a n/a designation. Maps were created using ArcGIS by ESRI version 10.5 using basemaps from www.arcgis.com and www150.statcan.gc.ca.

Provinces and public health authorities varied in the relative proportion of cat-exposed bats (Table 1 and Fig 1). For example, in Ontario, cat-associated submissions were 17%, compared to Alberta and British Columbia which had a higher proportion of submissions associated with cats at 52% and 60%, respectively. Lifestyle information could be inferred for 606 of the cat-associated submissions, of which 552 and 54 were confirmed outdoor or indoor cats, respectively.

There were 435 submitted bats that were positive for rabies (Table 1 and S1 Data), with 85 and 62 of these cases having exposure to cats and dogs, respectively. Twenty-one rabies-positive bats had a history of interaction with an outdoor cat, with a calculated prevalence in this group of 4.1% (95% CI: 2.5–6.2%). The prevalence of rabies-positive bats interacting with indoor cats was 1.9% (95% CI: 0.05–10.4%), with only one record of a rabies-positive bat being caught by an indoor cat. The rabies prevalence of bats interacting with cats of unspecified

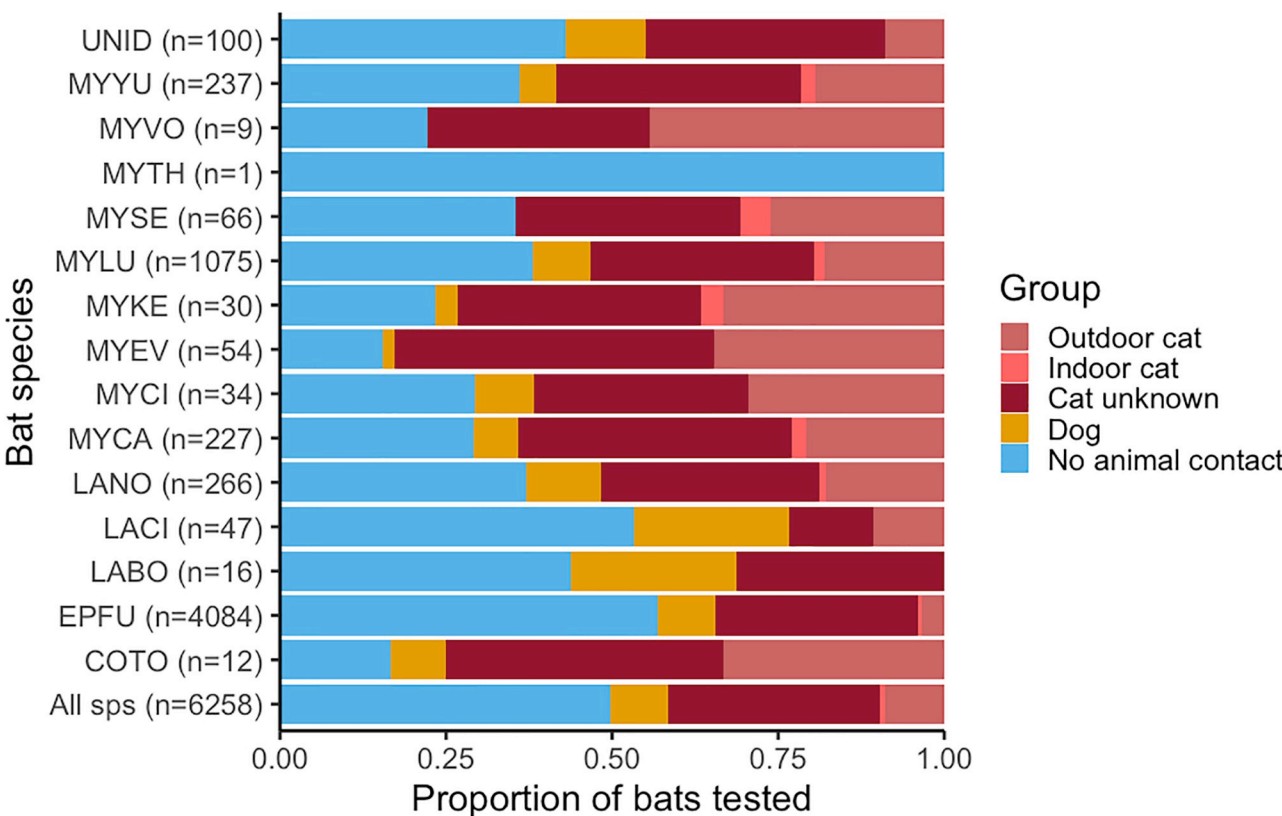

**Fig 2. Proportion of bat submissions across all provinces from 2014–2020 that were associated with outdoor cats, indoor cats, cats of unspecified lifestyle, dogs and no domestic animal.** Bat species codes are provided in the text.

lifestyles was 3.3% (95% CI: 2.5–4.3%). For the bats interacting with free-roaming cats, 7.4% (N = 41) of the samples were unfit for sampling, compared to 5.5% in the indoor (N = 3) or unknown cats (N = 109). Dog-associated bats had a rabies prevalence of 12.8% (95% CI: 9.9–16.1%) and an unfit sample submission rate of 10.7% (N = 58). Rabies prevalence was 9.7% (95% CI: 8.7–10.9%) in submitted bats with no history of cat or dog association with an unfit sample submission rate of 5.8% (N = 180).

The model with the highest support for predicting the submission of a rabies-positive bat was a model that included fixed effects of season, domestic species exposure and random effects of year, province and bat species (Table 2). Based on beta coefficients from this top

**Table 2. Akaike's Information Criterion (AIC) model selection results for evaluating the factors associated with detecting rabies-positive bat submitted to CFIA 2014–2020.**

| Model structure | AIC | ΔAIC | df |
|---|---|---|---|
| dom exp + season + (1\|year) + (1\|province) + (1\|bat.sps) | 2800.7 | 0 | 11 |
| season + (1\|year) + (1\|province) + (1\|bat.sps) | 2886.8 | 86.1 | 7 |
| (1\|year) + (1\|province) + (1\|bat.sps) | 2992.8 | 192.1 | 3 |

ΔAIC = change in AIC relative to the top model and df = the degrees of freedom.

Fixed effects: dom exp = exposure to domestic animals, season = season. Random effects: 1|year = year of submission, 1|province = province of submission, 1|bat.sps = bat species

model, we calculated that given that a bat was encountered, the probability that an outdoor cat (β = -2.60 ± 0.42) encountered a rabies-positive bat was 6.9%, while an indoor cat had a probability of 3.8% (β = -3.22 ± 1.06). The probability that a bat encountered by a dog or with no domestic animal exposure was rabies-positive was 20% (β = -1.37 ± 0.39) and 16% (β = -1.61 ± 0.38) respectively. Among the random effects, bat species identity accounted for 73.8% of the random variation in the model, while province and year accounted for 23.8% and 2.4%, respectively.

## Discussion

The first significant finding of this study was that almost half of the bats submitted to the CFIA laboratories had a history of an interaction with a cat, compared to less than 10% of bats being associated with dogs. In a previous study in Washington state, there was an even greater disparity, with 89% of bat submissions being cat-associated [29], suggesting that this pattern is not limited to Canada. Like cats, free-roaming dogs can severely impact wildlife populations through predation and disease [30]; however, in most communities in Canada and the United States, the free-roaming of dogs is not widely tolerated due to canine welfare and public safety concerns. Consequently, in North America, dog-bat interactions are more likely to be incidental and involve contained or controlled dogs. We propose that the near five-fold higher interaction rate between bats and cats compared to dogs is due to increased free-roaming rates of cats, which is also supported by the much higher submission rate of bats from free-roaming compared to indoor cats (91.1% versus 8.9% respectively). Although it is estimated that there are more owned cats than dogs in Canada (7.9 million cats and 5.9 million dogs [31]), the relative population size difference is insufficient to account for the significant disparity in bat-interaction rate between dogs and cats.

Significant predation impacts of cats on bats are evident from meta-analyses [18, 32–36], wildlife rehabilitation data [17, 37–39], and feral cat diet studies [40–42]. Even single cats can have substantial impacts. In New Zealand, one male cat killed more than 100 bats at a single natural roost in one week [43], and an owner of a female outdoor cat in Creston, British Columbia, reported that the cat killed 14 bats in a single evening (Craig pers. comm). Similarly, in this database, in less than one month, a single cat in Medicine Hat, Alberta, was responsible for nine submissions of the endangered little brown bat. Bats are especially vulnerable because many bat species are colonial, with large numbers of bats predictably exiting roosts at dusk or dawn, often through small access sites predisposing them to severe predatory losses. Adult mortality in bats will also have significant demographic consequences because bats are long-lived with low fecundity [44], with many populations undergoing decline [45, 46]. Therefore, we suggest that the conservation impact of free-roaming cats on Canadian bat populations warrants research attention and rapid intervention.

The second important result was the differences in rabies positivity among bats associated with dogs and indoor and free-roaming cats. Dogs had a higher rate of encountering rabies-positive bats, which agreed with our predictions that cats may be more likely to depredate healthy bats. Similar patterns were also found in rabies testing programs in Washington state [29]. Data from the BC provincial Animal Health Center also suggest cats are killing healthy bats; across 220 bat necropsies, there were no cat attack cases that were diagnosed with other health issues such as dehydration, emaciation, or pneumonia (Schofer pers. comm). Bats with clinical rabies are more likely to be disorientated and grounded and thus have an increased likelihood of being incidentally intercepted by a dog or human [29, 47].

Although we found that rabies prevalence in cat-associated bats was lower relative to dogs, this is offset by the 4.8-fold higher interaction rate of cats. Compared to indoor cats, free-

roaming cats had a ten-fold higher interaction rate with bats and double the probability of encountering a rabies-positive bat. In fact, in this dataset, there were five case submissions where a cat was explicitly described as bringing a bat into the house that was subsequently determined to be rabies-positive. Overall, these findings support our hypothesis that free-roaming cats facilitate the transfer of zoonotic disease across the human-wildlife interface. Indeed, free-roaming cats likely play an under-appreciated role in cryptic or even recognized human rabies exposures by bringing rabies-positive bats into close proximity to humans. For example, the most common rabies virus variants for cryptic human exposures are attributed to silver-haired and tricolored bats, neither of which commonly roost in homes [48–50].

Due to increased interactions with rabies reservoir species or interactions with feral cats, free-roaming owned cats have an amplified risk of becoming infected by rabies themselves [11, 19, 51, 52], with bat predation being an additional route. Bat rabies variants have been recovered from rabies-infected cats [21, 53], and cats experimentally infected with bat-variants were susceptible [54]. In the United States, cats are the most common rabies-positive domestic animal [21], and in Pennsylvania, cats pose 2.5 times the relative risk of human rabies exposure compared to bats [51]. Although studies suggest that cats generate a protective antibody titre more effectively than dogs [55], reported vaccination rates of owned cats (54%, [56, 57]) and feral cats (1.6%) [52, 58]) are wholly insufficient to be the only preventative approach. Limiting free-roaming was an essential intervention in the eradication of enzootic canine rabies in North America [59], and would be a highly impactful approach for cats as well. Although poorly known, free-roaming rates of owned cats in Canada are estimated to range from 40–70% [14]. Public attitudes towards free-roaming in cats are likely influenced by municipal by-laws, climate, socioeconomic status and awareness of the negative feline welfare, zoonotic and conservation consequences associated with free-roaming. Spatial patterns present in this study suggest there is substantial geographic variation in the extent of free-roaming in cats, but the lack of consistent reporting in rabies submission reports precludes analysis. Public health professionals need data on the broad and fine-scale dynamics of free-roaming cat populations to implement community-specific interventions [60] in order to reduce the community health burden and financial costs associated with feline-associated zoonoses [16, 61].

We acknowledge several caveats in this study. First, we cannot attribute different bat encounter rates between cats and dogs solely to free-roaming propensities since this specific information is not routinely collected. Documenting the presence of free-roaming pets in any rabies submission case is crucial epidemiological information that is directly pertinent to rabies prevention programs. For example, very different interventions are needed to reduce exposures to free-roaming cats bringing prey into the property versus encountering a bat that has become entrapped within a house.

Second, we acknowledge that susceptibility to rabies infection varies among domestic and wild animals, including bat species, which is further dependent on the rabies virus variant, virus dosage load and the site of inoculation [54, 62–65]. Although variability in susceptibility may alter patterns of direct rabies transmission from domestic species, existing data are insufficient to justify any alteration in public health recommendations for feline vaccination, and domestic animal interactions with bats should be avoided for both public health and conservation reasons. Furthermore, domestic species susceptibility does not affect the risk associated with free-roaming animals bringing a rabies reservoir species into direct contact with humans.

Third, rabies-testing data only approximates the conservation and rabies risks since this dataset only captures a small fraction of the bats killed by owned free-roaming cats. Only 20% of wildlife killed by a cat is returned to its owner [10, 66], such that the majority of predation events are unrecognized. Applied to the dataset in this study, although we would not expect the rabies prevalence to change, the much larger number of potential hunting events means

that the cumulative rabies risk and conservation impacts of any free-roaming cats would be five times higher than what is presented here. In addition, rabies bat submissions are unlikely to include predation due to feral cats, whose wildlife interactions are thought to be ten-fold higher in frequency than free-roaming owned cats [9, 10]. Feral cat predation of bats is a significant conservation threat [17, 18, 40, 43], and the direct rabies risk presented by feral cats is a recognized public health concern [51, 52]. However, because feral cats, by definition, have reduced direct human interactions, feral cats would be expected to have a more limited role in indirect rabies transmission through bringing bats directly to humans.

## Conclusion

Public health practitioners, physicians and veterinarians are at the forefront of investigating disease transmission pathways across wildlife-domestic animal-human interfaces. Free-roaming cats have an underappreciated role at this interface [19, 20], with the public being generally unaware of the zoonotic disease risk that free-roaming owned and feral cats pose for multiple pathogens [11, 19, 67, 68]. Accurate framing of public health messaging is crucial for preemptively avoiding interactions that are detrimental for humans, domestic animals and bats. Globally, public misinformation has led to the persecution of bats [69], leading to the loss of ecosystem services that ironically benefits human health and food security [20, 70]. In this database, there were multiple cases where bats were killed by members of the public using inhumane techniques, reflecting irrational fear and inadequate public appreciation for relative rabies risk, the essential ecological role of bats, species-specific legislative protection and their capacity for pain and suffering.

Considering the substantial conservation crises facing bats [46], their provision of billions of dollars of ecosystem services [20, 70] and the human and financial costs of zoonotic disease spillover [19], there are no ethical or financial justifications for any complacency regarding free-roaming cat predation on wildlife. Furthermore, due to the endangered status of the little brown bat (*M. lucifugus*), the northern myotis (*M. septentrionalis*) and the tri-colored bat (*P. subflavus*) on Canada's Species-at-Risk list [71], it is imperative that judicious policies are adopted to avoid preventable bat euthanasia wherever possible.

Limiting the unsupervised free-roaming of cats through investing in promoting responsible pet ownership, no-roaming by-laws, and enclosed feral cat sanctuaries should be viewed as cost-effective public health interventions; interventions that also simultaneously benefit feline and wildlife welfare and conservation. Although the zoonotic risk of free-roaming in cats is often overlooked or omitted in public health policies [11], it is a poignant example of the connection between public health and wildlife conservation, which is a foundational principle of the global One Health movement.

## Supporting information

**S1 Data. Raw dataset for bats submitted for rabies testing to the Canadian Food Inspection Agency's Centre of Expertise for Rabies from 2014 to 2020.**
(XLSX)

## Acknowledgments

Special thanks to the staff of the CFIA Rabies Laboratories (Ottawa and Lethbridge) who conducted the diagnostic testing and to Juliet Craig, Rachel Avilla, Delaney Schofer, Carol Kelly, Kate Rugroden, Lisa Tretiak, Amanda Lollar, and John Saremba for assistance and discussion on cat-associated mortalities on bats. Jared Hobbs provided the photo of the Silver-haired bat.

We thank Megan Griffiths, Catherine Brisson and Leigh Anne Isaac for comments and suggestions that greatly improved this manuscript.

## Author Contributions

**Conceptualization:** Amy G. Wilson, Scott Wilson, Tanya M. J. Luszcz.

**Data curation:** Amy G. Wilson, Christine Fehlner-Gardiner, Karra N. Pierce, Glenna F. McGregor.

**Formal analysis:** Amy G. Wilson, Scott Wilson, Catalina González.

**Funding acquisition:** Tanya M. J. Luszcz.

**Methodology:** Christine Fehlner-Gardiner, Glenna F. McGregor.

**Project administration:** Christine Fehlner-Gardiner.

**Validation:** Christine Fehlner-Gardiner.

**Visualization:** Catalina González.

**Writing – original draft:** Amy G. Wilson.

**Writing – review & editing:** Amy G. Wilson, Christine Fehlner-Gardiner, Scott Wilson, Karra N. Pierce, Glenna F. McGregor, Catalina González, Tanya M. J. Luszcz.

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
