## [Decision Letter · Decision Letter 0]

8 Feb 2022

PGPH-D-22-00039

Assessing the extent and public health impact of bat predation by domestic animals using data from a rabies passive surveillance program

Dear Dr. Amy Wilson,

Thank you for submitting your manuscript to PLOS Global Public Health. After careful consideration, we feel that it has merit but does not fully meet PLOS Global Public Health’s publication criteria as it currently stands. Therefore, we invite you to submit a revised version of the manuscript that addresses the points raised during the review process.

We look forward to receiving your revised manuscript.

Kind regards,

Muhammad Asaduzzaman, MD MPH MPhil

Academic Editor

Journal Requirements:

1. Please update your Competing Interests statement. If you have no competing interests to declare, please state: “The authors have declared that no competing interests exist.”

2. Please provide separate figure files in .tif or .eps format only and ensure that all files are under our size limit of 20MB.

3. Please provide a complete Data Availability Statement in the submission form. If your research concerns only data provided within your submission, please write “All data are in the manuscript and/or supporting information files.” as your Data Availability Statement.

4. msdata.xlsx is currently uploaded as file type “Other”, which is not viewable by the reviewers.  Please change these to “Supporting Information” and include a legend in the manuscript if you wish them to be included in review.

5. We noticed that you used “data not shown”/"unpublished data" in the manuscript. We do not allow these references, as the PLOS data access policy requires that all data be either published with the manuscript or made available in a publicly accessible database. Please either remove these references, or amend the supplementary material to include the referenced data.

6. Please provide us with a direct link to the base layer of the map used in Figure 1 and ensure this location is also included in the figure legend. 

Please note that, because all PLOS articles are published under a CC BY license (creativecommons.org/licenses/by/4.0/), we cannot publish proprietary maps such as Google Maps, Mapquest or other copyrighted maps. If your map was obtained from a copyrighted source please amend the figure so that the base map used is from an openly available source.

Please note that only the following CC BY licences are compatible with PLOS licence: CC BY 4.0, CC BY 2.0  and CC BY 3.0, meanwhile such licences as CC BY-ND 3.0 and others are not compatible due to additional restrictions. If you are unsure whether you can use a map or not, please do reach out and we will be able to help you. 

The following websites are good examples of where you can source open access or public domain maps:

Additional Editor Comments (if provided):

I appreciate the novel approach to observe public health impact of bat predation by domestic animals. It would be comprehensive to add few more tables or figures for better data visualization and statistical analysis. Looking forward to see the revised version.

Reviewers' comments:

Reviewer's Responses to Questions

**Comments to the Author**

1. Does this manuscript meet PLOS Global Public Health’s publication criteria? Is the manuscript technically sound, and do the data support the conclusions? The manuscript must describe methodologically and ethically rigorous research with conclusions that are appropriately drawn based on the data presented.

Reviewer #1: Yes

Reviewer #2: Yes

2. Has the statistical analysis been performed appropriately and rigorously?

Reviewer #1: Yes

Reviewer #2: N/A

3. Have the authors made all data underlying the findings in their manuscript fully available (please refer to the Data Availability Statement at the start of the manuscript PDF file)?

Reviewer #1: Yes

Reviewer #2: Yes

4. Is the manuscript presented in an intelligible fashion and written in standard English?

Reviewer #1: Yes

Reviewer #2: Yes

5. Review Comments to the Author

Reviewer #1: In this paper, the authors use existing data concerning encounters between wild bats and cats, dogs, and humans from the CFIA, to evaluate the comparative exposure risks from these groups. The authors’ overall assessment is that the greatest risk of rabies exposure comes from free-roaming outdoor cats, and that given this exposure risk, policies to reduce free-roaming of outdoor cats would be a valuable public health and conservation measure. This paper addresses the important and underappreciated role in rabies transmission that cats can have, given that public perception of rabies is largely canine related, and although of limited scope, contributes a Canadian case study to the body of evidence pointing towards the risks of outdoor cat depredation of bats. Whilst the data has its limitations, specifically concerning inevitable under-reporting of encounters, I find the paper to be clear and succinct, with a reasonable discussion of the data and its limits.

Overall, I support the publication of this interesting manuscript. Below are some points that the authors should consider in order to clarify the manuscript.

Major points:

In general, the paper would benefit from a clear breakdown of the data involved (in the form of a table or figure) that can be referred to throughout. I believe that all of the data used in the paper is available, but not necessarily in the most comprehensive format.

Line 153: ‘Overall, submitted bats had contact with cats for 41.5% of all cases (n=2599), while 8.7% of bats had contact with dogs (n=542) (Figures 1 & 2). Cases with no known human contact comprised 49.5% of the submissions (n=3097).’ This differs from the abstract which states on line 29 ‘… 49.5% had no animal exposure’. Please clarify the nature of this 49.5% of submissions, as at present I’m unclear as to where these samples originate. This section of the results would also benefit from the 0.3% of other animals mentioned in the abstract, so that the full 100% of sample origins is apparent.

Minor points:

For the paragraph including the above points (beginning line 153) I find the order in which the data are presented somewhat disjointed, as the results jump between the contact exposure history of submitted samples, to the rabies results, then back to the exposure histories. I would recommend that the initial paragraph focus on exposure history, and the second (currently line 173) contains all of the rabies positives data.

Figure 1: If possible, this figure would benefit from the legend not being faded, so that the colours match those on the map. I would also appreciate the labelling of the map with the abbreviations for each province as in Table 1.

This figure and the text on line 157 show differing proportions of cat-associated submissions, would the authors care to comment on the possible reasons for such variation? Do the proportion of indoor to outdoor cats differ between these regions? If this data is available, it might be interesting to compare.

Figure 2: A final bar at the top or bottom of this figure with the cumulative results for all bat species would be useful. Perhaps the proportion of each bat species that tested positive for rabies (not per exposure group) could also be incorporated in some way, for a visual representation of which bat species to have the highest rabies prevalence.

Line 174: ‘There were 21 rabies-positive bats submitted after interaction with an outdoor cat, with a calculated prevalence in this group of 4.1%’ – does this calculation use the aforementioned 552 outdoor cat submissions? Since this prevalence does not quite match the one presented in the paper, I assume that this id due to several of the outdoor cat encounter submissions being unsuitable for testing. As such, a breakdown of the data including exposure group, total suitable for testing, and positive tests would be useful (as mentioned above).

Line 193: Are the models found in Table 2 all of the models that were tested, or the top three models? Were any other combinations of fixed and random effects considered?

Line 276: The authors cite that a predicted 20% of wildlife caught by free roaming cats is returned to owners, and thus available for submission for rabies testing. Assuming that this figure is accurate, could the authors speculate on the true risk of rabies exposure to outdoor cats?

Line 277: It may be pertinent to mention that although feral cats will have a far greater risk of interacting with rabid bats, and therefore are a large ecological concern, interactions with humans will be far less common than owned cats. Could also mention however, that owned free-roaming cats may interact with the more at-risk feral cats, even if their owners do not.

Very minor:

Line 76: ‘there are multiple terrestrial mammalian reservoirs for rabies, such as dogs, fox, skunk, raccoon, and bats’ – I would not consider bats to be terrestrial reservoirs.

Line 192: I believe that the table description should read ‘degrees of freedom’ as supposed to ‘degree of freedoms’.

Reviewer #2: This paper presents an interesting examination of the extent and public health impact of bat predation by domestic animals using data from a rabies passive surveillance program. The authors found rabies prevalence was higher in bats found by dogs (12.8%) and humans (9.7%) followed by outdoor cats (4.1%) and indoor cats (1.9%). Although there was lower rabies prevalence in bats encountered by cats compared to dogs and humans, the much higher number of cat-bat interactions results in a greater overall rabies exposure risk to humans from free-roaming outdoor cats. In this case, the authors suggested that preventing free-roaming in cats is an underutilized public health recommendation. Overall, it is a novel and good study for which I have several major comments. Please find them below.

The manuscript would benefit from a clear articulation of the hypothesis that underlies the work. This should be followed up by specific predictions that can be tested using the data collected during the research. The statement of objectives in this Introduction section between line 86 to line 100 does not do an effective job of setting the stage for this research. It does not move the work out of a more descriptive context.

In this study, these results about the differences in the numbers of bats tested between provinces, between bat species and between domestic animals contacting with bats were mostly descriptive. It is possible to test these differences by some statistical analysis.

Although the authors considered bat species as a random factor in generalized linear mixed models, every bat species may has different susceptibility to the rabies virus. In this case, it seems to be necessary to consider phylogenetic relationships of the 14 species of bats during the statistical analysis.

In this study, the different domestic animals had the different rate of encountering rabies positive bats, but the risk of rabies infection in domestic animals is difficult to determine because of the following two reasons. On the one hand, data of rabies prevalence in different domestic animals was lacking. On the other hand, every domestic animal may has different susceptibility to the rabies virus. Thus, the factors should be considered and discussed in making intervention recommendation for public health.

In Figure 1, latitude, longitude, north and scale should be marked.

6. PLOS authors have the option to publish the peer review history of their article (what does this mean?). If published, this will include your full peer review and any attached files.

**Do you want your identity to be public for this peer review?** For information about this choice, including consent withdrawal, please see our Privacy Policy.

Reviewer #1: **Yes: **Megan Griffiths

Reviewer #2: No

---

## [Editor Report · Decision Letter 1]

11 Apr 2022

Assessing the extent and public health impact of bat predation by domestic animals using data from a rabies passive surveillance program

PGPH-D-22-00039R1

Dear Amy Wilson,

We are pleased to inform you that your manuscript 'Assessing the extent and public health impact of bat predation by domestic animals using data from a rabies passive surveillance program' has been provisionally accepted for publication in PLOS Global Public Health.

Best regards,

Muhammad Asaduzzaman, MD MPH MPhil

Academic Editor